

# Rainfall feature extraction using cluster analysis and its application on displacement prediction for a cleavage-parallel landslide in the Three-Gorges Reservoir area

**Y. Liu[1], L. Liu[2]**

[1]Faculty of Mechanical and Electronic Information, China University of Geosciences, Wuhan, 430074, China
[2]Department of Civil & Environmental Engineering, University of Connecticut, Storrs, CT 06269-3037, USA

*Correspondence to:* L. Liu (Lanbo.Liu@UConn.edu)

**Abstract.** Rainfall is one of the most important factors controlling landslide deformation and failure. State-of-art rainfall data collection is a common practice in modern landslide research world-wide. Nevertheless, in spite of the availability of high-accuracy rainfall data, it is not a trivial process to diligently incorporate rainfall data in predicting landslide stability due to large quantity, tremendous variety, and wealth multiplicity of rainfall data. Up to date, most of the pre-process procedure of rainfall data only use mean value, maxima and minima to characterize the rainfall feature. This practice significantly overlooks many important and intrinsic features contained in the rainfall data. In this paper, we employ cluster analysis (CA)-based feature analysis to rainfall data for rainfall feature extraction. This method effectively extracts the most significant features of a rainfall sequence and greatly reduced rainfall data quantities. Meanwhile it also improves rainfall data availability.

For showing the efficiency of using the CA characterized rainfall data input, we present three schemes to input rainfall data in back propagation (BP) neural network to forecast landslide displacement. These three schemes are: the original daily rainfall, monthly rainfall, and CA extracted rainfall features. Based on the examination of the root mean square error (RMSE) of the landslide displacement prediction, it is clear that using the CA extracted rainfall features input significantly improve the ability of accurate landslide prediction.

## 1 Introduction

Landslides are one of the major geological hazards cause major life loss and socio-economic disruption each year world-widely. An early warning system for potential landslide in steep mountainous area with landslide-prone segments is an effective approach to avoid property damage and casualties. To make the early warning systems function effectively and reliably, information on the behaviour of the landslide, including the sliding mechanics, the potential triggering mechanism, and the critical precursors of slope stability for issuing emergency warnings are the major parameters to be sought. The most critical parameters for early warning output are creep velocity, displacement magnitude, and instability prediction (Sassa et al. 2009).

Rainfall is not only a crucial index of landslide analysis but also a significant factor in triggering landslides. At present time, rainfall data collected is very accurate and we can perform statistical analysis based on daily or even real-time data. However, considering that rainfall data becomes more accurate and thus data volume becomes



bigger, it is difficult to directly use them in landslide analysis. In previous research work, Cepe da et al. (2010) applied rainfall data to estimation of landslides probability in spatial prediction. Rossi et al. (2012) discussed the

rainfall threshold of regional landslide in spatial prediction. Melillo et al. (2014) proposed an algorithm calculating rainfall threshold for different landslides. Many people have also studied triggering mechanism between rainfall and landslide, (e.g., Lee et al. 2014; Li and He 2012). These researches are aimed at the rainfall in a particular landslide, and the relationship between rainfall threshold triggered by landslide and the probability of landslide, or rainfall probability in regional landslide and the probability of landslide, with no processing of data. With the recognition of

the importance of rainfall data growing, attaching greater significance to the information contained in data, some scholars have begun to study the rainfall data itself. Saito et al. (2010) divided rainfall of landslide in shallow condition into two types: short-cycle, high-intensity (SH) and long-time, low-intensity (LL), putting forward the fact that different types influenced landslide differently. These studies have shown the fact that rainfall data is worthy of digging deeply information they contain to disclose the effects of different rainfall types on landslides.

More innovative data-processing and information fusion methods such as Feature Analysis, Feature Extraction etc., have emerged and been applied in the processing of landslide monitoring data in recent years. These new approaches can be classified into two categories. The first one is to use the feature extraction of radar detection data to forecast and analyse landslide. For example, Wang et al. (2010) applied airborne-radar data to topographic patterns extraction, and predictions about geological disasters such as landslides. The other category is to acquire

relevant information and deformation of landslide through feature extraction of remote sensing images of landslide. For example, Kurtz et al. (2014) acquired the boundary and configuration about landslide through the feature extraction of very high resolution images. Through studies like this, we can draw a conclusion that the feature extraction methods of landslide are mainly concentrated on the processing of radar data and remote sensing data. Very few studies involved analysis of rainfall data in monitoring landslide.

According to previous research work, rainfall data plays a very crucial role in landslide deformation and failures, especially in the cases of rainfall-landslide type. Utilizing some methods processing data, such as quantitative and extreme methods, are not capable to dig out the important information contained in data. Although recently scholars have started to categorize data and conduct information mining for rainfall data, there is a lack of substantial researches in this direction. In this paper, we performed feature extraction method to the rainfall data which is

categorized as clustering analysis. With this approach the computation stress is greatly reduced; meanwhile, critical information can be extracted from the data. Finally, this approach is applied and validated to a data set acquired at a cleavage-parallel landslide in the Three-Gorges Reservoir area.

        The rest of this paper is organized as follows. First, the feature analysis of rainfall data, the relationship between rainfall and evaporation capacity, as well as their influences on rainfall and landslides are discussed. Characteristic

indices of rainfall, such as rainfall quantity, duration, and the number of raining days in a given period of time will be introduced. The explanations of how to use clustering analysis to categorize rainfall data, including selection of feature and weight analysis of data are followed. Then, the basic of Clustering Analysis is briefly introduced. Finally, application of feature analysis and feature extraction of rainfall data fora bedding landslide monitoring in The Three-Gorges area between July 2003 and December 2008 is presented as a case study. Land slide displacement



prediction using BP neural network for the rainfall input in the form of raw data, monthly rainfall, and feature extracted rainfall are compared. The final results demonstrated that the one using featured rainfall has the best forecasting with root mean square error (RMSE).

## 2. Methodology

For the cleavage-parallel landslide, rainfall is a very important factor. In the existing studies, simple numerical
methods, such as the cumulative rainfall method (Bi et al. 2004), the average annual rainfall method (Liao et al. 2011), or the 1-day, 3-day, or 7-day maximum rainfall method (Huang 2011) were used to extract rainfall features. These works overlooked some of the important information contained in the rainfall data. It is usually admitted that continuous and heavy rainfalls are necessary conditions in triggering landslides in qualitative analysis; however, intermittent rainfall or sporadic rainfall can also have certain non-negligible influence on the stability of landslide.
There are other factors in rainfall affecting landslides, including evaporation, volume, number of times, and duration. These factors are discussed below in details.

### 2.1 The relationship between rainfall and evaporation

In the studies of rainfall effect on landslides, evaporation is a factor that cannot be simply ignored. The monthly average of evaporations is highly variable, and the changes can be very dramatic. For example, in the Three-Gorges
Reservoir area, the evaporation is only about 1 mm/d in winter and spring, but may reach 7 mm/d in a hot summer day. When the evaporation is high and the rainfall is low, rainfall has very little effects on landslide. Usually we would deem rainfall volume less than evaporation invalid in this study. In other words, we cannot talk about rainfall alone without taking into the count of the evaporation.

     In this study, we will calculate the average daily evaporation in every month. If the daily rainfall is greater than
the average daily evaporation in the month, we would consider it is valid rainfall data. If the daily rainfall is less than or equal to the average daily evaporation, we would consider it is invalid rainfall data and the actual rainfall data for that day will be deemed zero.

### 2.2 Statistics of rainfall by times

Up to date, most studies carry out statistics analysis of rainfall based on precipitation per month or per day, or select
extreme values in a month or in a few days for landslide analysis. Such statistics do not consider different rainfall types, and it is hard to show the features of rainfalls. In our study, we calculate the number of raining days. When the rainfall volume is less than or equal to the evaporation, we set effective rainfall to zero. According to the situation of rainfall and slope, we set a threshold N (with N=1, 2, 3). If there was no rainfall in the following N days after the first rainfall occurs, we count it as one time rainfall. We use clustering analysis to categorize all the data
after counting the rainfall times. By doing so, we are able to extract the features of each type, and conduct analysis in a more accurate way.





### 2.3 The features of the rainfall data: rainfall volume, rainfall duration and rainfall time

We need take the effects of different rainfall types on the landslides into account when we determine the factors in categorization. Saito et al. (2010) considered the amount of rainfall and rainfall time for the purpose of categorization. In a qualitative analysis, these two factors are usually considered. Based on previous statistics, we emphasize on rainfall duration which can distinguish continuous rainfall from intermittent rainfall. These two different types of rainfall could have different effects on landslide analysis. In this study, we categorize rainfall based on these factors: rainfall volume, rainfall duration and rainfall time.

Rainfall volume is an important index in categorization. In this research, we select the average daily rainfall volume, which is the rainfall volume divided by the number of days the particular rainfall lasts. Based on our comparative study, the average daily rainfall volume represents the rainfall intensity better and thus differentiates the strong rainfall from continuous rainfall with less ambiguity. The second index we have chosen is rainfall lasting days. It is an important index as it represents both the rainfall volume and the rainfall duration. The third index is the proportion of the raining days in the total number of rain days, which is a crucial index to distinguish continuous rainfall from intermittent rainfall. In addition, since we use millimetre as the measurement unit, the range of rainfall volume data will be (0, 80), the scope of raining days being (0, 6) and (raining time, duration) we need to scale the data to warrant that they are on the same quantitative level, through multiplication by particular coefficients. This will make sure high cohesion and low coupling among the data after categorization. After categorization, we select each kind of rainfall as a particular feature and extract the data, using the BP neural network to demonstrate the effectiveness of feature extraction.

Based on the feature analysis, we will need to categorize rainfall data. This paper adopts the clustering analysis method. We use the K-means algorithm to categorize rainfall data and we follow the rule of cluster analysis: high cohesion and low coupling in the analysis process. We try to ensure maximum similarity in same types and maximum dissimilarity in different types after categorization. We need to place those important features in prominent positions in selections of parameters and weight, in order to distinguish different types of rainfall data. First, a brief summary of the K-means clustering algorithm is presented below.

### 2.4 Clustering Analysis using the K-means clustering algorithm

Currently, K-means algorithm is the most widely used clustering algorithm. The basic idea of this algorithm is to use iteration to search K clusters and we can obtain minimum overall error using the mean of a set of K clusters to represent the corresponding samples. The algorithm is simple and fast to converge. The process follows the order using k-means algorithm. Firstly, we should select k objects as the initial centroid of k class. Then we categorize them according to the distance between the centroid and other objects. The centroid is then adjusted accordingly if there are new objects participating in the process. The process is repeated until the squared error converges.

$$J = \sum_{j=1}^{k} \sum_{i=1}^{n} \left\| x_i^{(j)} - c_j \right\|^2 \tag{1}$$

Where J is the squared error of all objects, $x_i$ represent the data points, $c_j$ represents the centroid the j-th cluster.



The biggest drawback of K-means algorithm is that the choice of initial point greatly affects the final results. Different choice of initial points will directly affect the accuracy of categorization. In selecting the initial points, we need to follow the principle of the largest dissimilarity, which means the less similar the initial centroid are, the more accurate the subsequent division will be. Also the appropriate approach to choose initial points can accelerate the convergence speed of the algorithm. We will design the selection plan in this paper, selecting k initial points in order to make quadratic sum of distance of k initial points the largest.

$$L = \sum_{i=1}^{k} \sum_{j=1}^{k} (x_i - x_j)^2 \tag{2}$$

We will select k nodes as the initial points in order to make L the largest in all selections.

## 3 Application to the Baishuihe Landslide field data in the Three-Gorges Reservoir

### 3.1 Geological background and data collection

The Baishuihe Landslide is located in the south bank of the Yangtze River, 56 km away from the Three-Gorges Dam (Fig. 1). The landslide is located in the relatively wide open area of Yangtze River valley. It is a single, north-facing, inclined cleavage-parallel slope. It is shaped like stairs to the Yangtze River. The rear edge (crown) is about 410 meters high from the front edge (toe). The toe is at the 140-m water level of the Yangtze River. Both the left and right flank sides are surrounded by a bedrock ridge and the dip angle is about 30 degrees. It is about 600-m long in sliding direction and 700-m wide laterally. The average depth of the landslide is about 30 meters, and the total volume is about 12.6 million cubic meters.

For monitoring the deformation of this landslide seven GPS monitoring benchmarks were built along three longitudinal profiles in the Yangtze River in June 2003 (the solid red squares with labels initiated by ZG). Later on in June 2005 four more GPS monitoring benchmarks have been added to the right part of the slide. There is a GPS reference point on each side of the flanks in the rock ridge. In order to better represent the landslide sliding incidence and verify our processing method of rainfall data, we select the displacement data of landslide monitoring point ZG93 as the experimental object for training and prediction of the neural network algorithm.

### 3.2 Feature analysis of rainfall data

We use the daily rainfall data from July 2003 to December 2008 (Fig. 2) in Zigui County, Hubei Province, China to conduct the analysis to seek the effects of rainfalls to landslide displacement of the Baishuihe Landslide.

As can be seen from Fig. 2, rainfall in this region mainly concentrates in the summer months from April to September, and the heaviest rainfalls happen in July. During this period of 5 years and 7 months, the highest daily rainfall volume is 81.8 mm, occurred in June 2006; while the longest continuous rainfall occurred in July 2008, lasted for more than 11 days.

Han et al. (2012) discussed the evaporation of Zigui area from 2001 to 2010. The average annual evaporation in Zigui County in the last decade is 937.0 mm, and the total evaporation between May and September is 668.5 mm, the monthly maximum is 187.8 mm, occurred in July. From October to next year's April, the total evaporation is only 269.1 mm. We take 4 mm/d as the daily average of evaporation for May, June, August, September, 6.26 mm/d





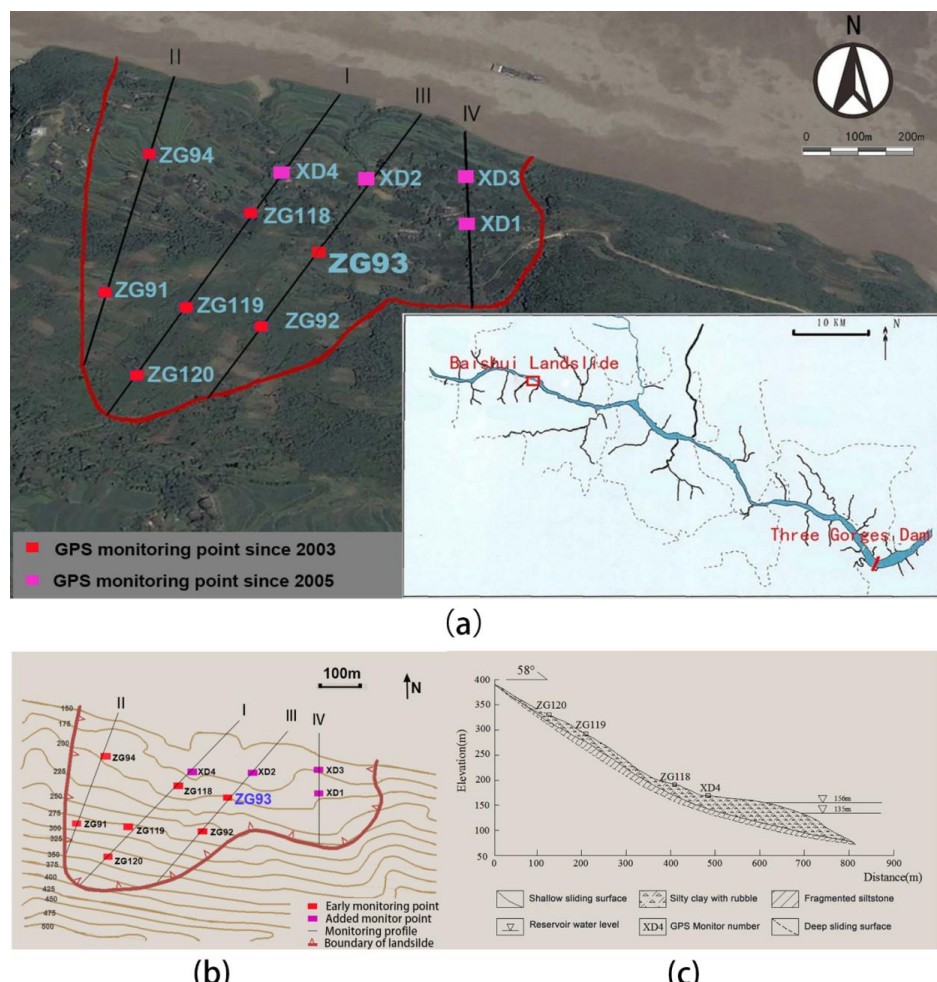

Fig. 1. The location of the Baishuihe Landslide (the west most red open square) in the Three-Gorges Reservoir area (a); The locations of the GPS benchmarks (the red and magenta solid squares) for displacement monitoring in the Baishuihe Landslide (b); and the vertical geological cross-section of the Baishuihe Landslide along Profile I (c)

for July, and 1.28 mm/d for October to April. By taking evaporation into account, when processing the rainfall data, if the daily average of evaporation is greater than the rainfall volume of a particular day, we consider the rainfall volume of that day to be zero. In other words, if rainfall volume is less than evaporation, it is deemed invalid rainfall. Only when the rainfall volume is higher than daily evaporation, the actual rainfall volume is used for data processing.

In this analysis, we set interval threshold of rainfall N equals to 2; that is to say, if there is no effective rainfall for 2 days, we consider the rainfall ends. If there is only one day without rainfall since the first raining day, we consider the rainfall is not over yet. The rain is over until there is no effective rainfall for 2 days. Based on this premise, the

170




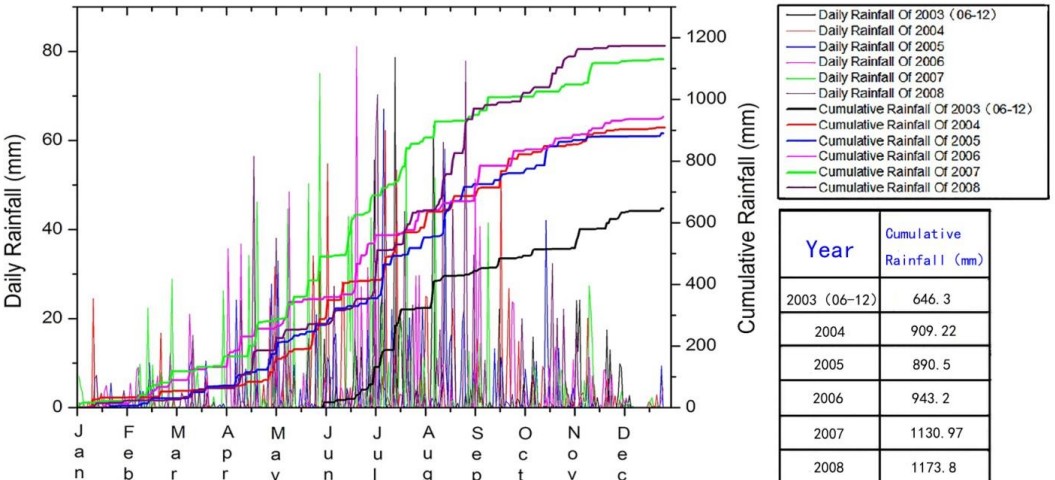

Fig. 2. Annual rainfall data in terms of daily and cumulative precipitations for the period of January 2003 to
December 2008

total number of rainfalls is 211 from July 2003 to December 2008, most of which is single rain-day, accounting for
112 events. More-than-one-day rainfalls account for 99 events.

### 3.3 Feature extraction of Rainfall data and Categorization results

After we sample the rainfall data based on the total number of rainfall events, we calculate average daily rainfall
indices each rainfall event. To ensure the data of these three indices be on the same order of magnitude, the three
features extracted are individually multiplied by a coefficient. The first index is the average daily rainfall volume r
defined as:

$$r = \frac{R}{d}p_1 \qquad (3)$$

Where R is the total volume of a rainfall event, d is the number of raining days in this rainfall event, $p_1$ is a
coefficient equals to 0.1. The measuring unit is millimeter.

The second index is the number of days of rainfall d. In our sample, its range is 1-6. So we will use the original
data without scaling.

The third index is the ratio of rainfall days over the continuous days T:

$$T = \frac{d}{D}p_2 \qquad (4)$$

Where d is the number of raining days, D is the total number of days during the particular rainfall event, and $p_2$ is a
scaling coefficient. According to our test, we can reach an optimal point of maximum cohesion and minimum
coupling effect when $p_2$ is set to be 11.





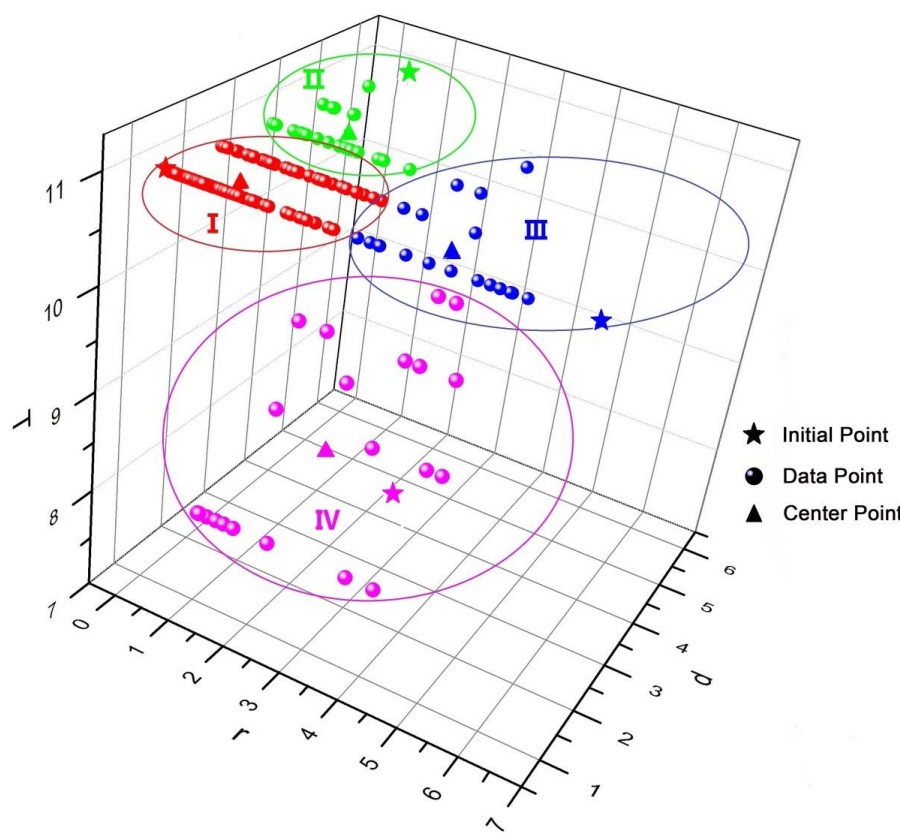

Fig. 3. The classified rainfall types based on cluster analysis: I: sporadic rainfall; II: long-duration rainfall; III: short-duration storms; and IV: long-duration intermittent rainfall

Using the K-means clustering algorithm to calculate the parameters r, d and T for each of these 211 rainfall events, we can characterize the rainfall events into four clusters, as shown in Fig. 3

To reduce the number of iterations and improve the clustering performance, four points are selected as the initial clustering centers based on the principle of maximum dissimilarity. These four initial points are C1=(0.13, 1, 11), C2=(0.52, 6, 11), C3=(7.51, 1, 11), C4=(2.45, 4, 7.33), respectively (see Fig. 3). We represent these points in the form of (r, d, T).

In the clustering process a new sample $x_i$ is added each time and use $M = \sqrt{\sum_{j=1}^{3}(x_{ij} - C_{ij})}$ to calculate the distance between this point and the four cluster centers. Based on the minimum distance principle this new sample is assigned to the closest cluster. Add the samples sequentially to exhaust these 211 samples; and each of rainfall samples must belong to one of these four clusters. Next, update the clusters centers by the formula $C_i = \frac{1}{n}\sum_{x \in C_i} x$, calculate distance between each sample and the new cluster centers, and re-cluster it according to the distances.



Repeat this cluster center updating process until the clustering becomes stable. Finally, four cluster centers: (1.00, 1.31, 11), (1.06, 3.42, 11), (4.23, 1.40, 11), and (1.69, 3.09, 7.99) are obtained. The above data are rounded to two decimals. There are 142 samples in the first cluster, 25 in the second cluster, 21 in the third cluster and 23 in the fourth cluster, as shown in Fig. 3. The first cluster (category) is characterized by low-rainfall, duration of 1-2 days, which are mainly the sporadic rainfalls (the red cluster in Fig. 3). The characteristics of the second type of rain are comparatively less volume, but with long duration and no interruption (the green cluster in Fig. 3). The third type of rainfall is characterized by short duration, usually 1-2 days, but the rainfall volume is very big (storms, the blue cluster in Fig. 3). Finally, the fourth type of rainfall is long duration with moderate rainfall volume and intermittent rainfall (the magenta cluster in Fig. 3).

After categorization, rainfall volume is still the most important factor in causing the variations of displacement in the cleavage-parallel landslide. Therefore, we use rainfall volume as the feature for extraction, taking the total rainfall volume in the same category as the feature of that particular category. In displacement prediction as described later in this paper, we conduct statistics on the rainfall volume per month of each type of rainfalls. For example, in the period between August 16 and September 15, 2008, there were five events of effective rainfall. The five samples were measured using our (r, d, T) set at (0.98, 2, 11), (3.39, 2, 11), (3.68, 3, 11), (3.43, 1, 11) and (1.07, 1, 11), respectively. Among this small sample set, there were 2 first-type rainfalls, 0 second-type rainfalls, 3 third-type rainfalls, and 0 fourth-type rainfalls. The total rainfall volumes were 30.3, 0, 212.5, and 0 mm for each type of the rainfall respectively. Using feature extraction, the feature vector for rainfall in that month would be (30.3, 0, 212.5, 0).

### 3.4 Prediction of landslide displacement with BP neural network

After the discussion of rainfall feature characterization and extraction with the clustering algorithm, we are ready to touch the major topic of the effect of rainfalls on landslide displacement. Using simple correlation just shown as Fig. 4, one can find that the connection between rainfall and landslide displacement at the Baishuihe site is quite obvious. Nevertheless, more closed and quantitative examination is needed to enable us reach more definitive conclusion of this causality.

To verify the effectiveness of feature extraction after using cluster analysis, we utilize BP neural network to forecast displacement for the follow three treatments of the rainfall data: 1) direct use of original rainfall data; 2) monthly average rainfall; and 3) rainfall data processed through cluster analysis and feature extraction. We use the rainfall data of the current month and the last month, along with the displacement of last month as the input to predict the displacement in the current month with BP neural network. We use only one hidden layer. And the node number is $n_1$ with: $n_1 = \sqrt{n+m} + a$; where n is the input layer node number, m is the output layer node number; and a is a constant, which is set to be 2 in this work.





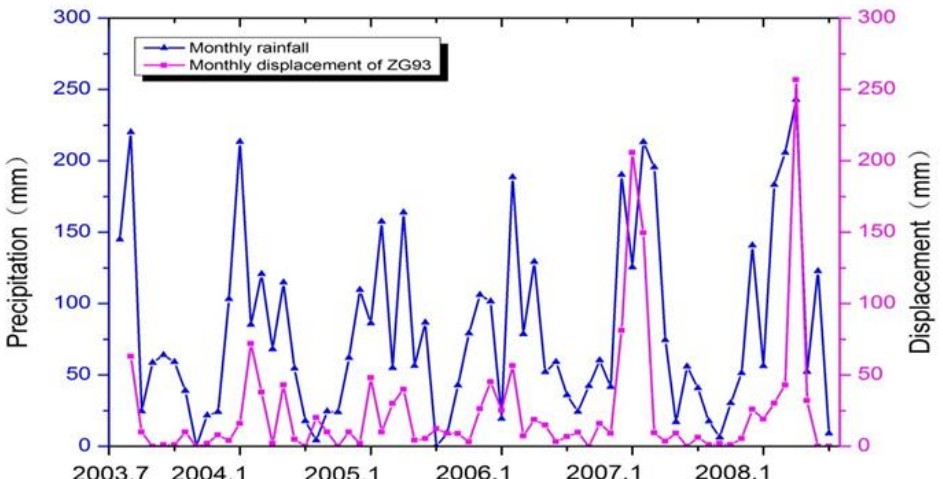

Fig. 4. The monthly rainfall in Zigui County and displacement recorded by GPS survey mark ZG93 at the Baishuihe site for the period of July 2003 to June 2008

First, we use rainfall data and displacement data between July 2003 and December 2005 to train the BP neural network. Then we use the trained neural network to predict displacement between January 2006 and December 2008. In the prediction process, once the prediction of the displacement of each month is finished, we use the newly obtained data to train the neural network again, and use the newly trained network for prediction of the displacement of next month. The prediction results are shown as Figs. 5 and 6; and the network structure(show as [input node number, hidden node number, output node number]); the errors of training; operation times of training and prediction by BP neural network is shown in Table 1.

Table 1: The root mean square error of training of and prediction by BP neural network

|  | Network structure | RMSE of training | Operation times of training | RMSE of prediction |
|---|---|---|---|---|
| Direct use of daily rainfall | [61,9,1] | 0.000455022 | 1.21*109 | 322.85 |
| Monthly average rainfall | [3,4,1] | 1.084612 | 2.33*107 | 407.68 |
| Rainfall w/ feature extraction | [9,5,1] | 0.002425925 | 1.40*108 | 261.93 |

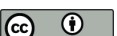


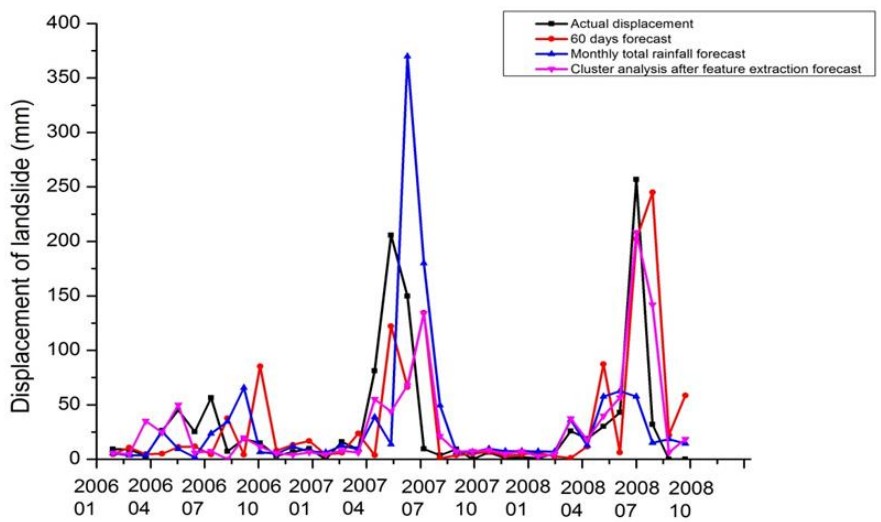

Fig.5. Landslide displacement prediction based on the 3 methods for accounting for the rainfall input

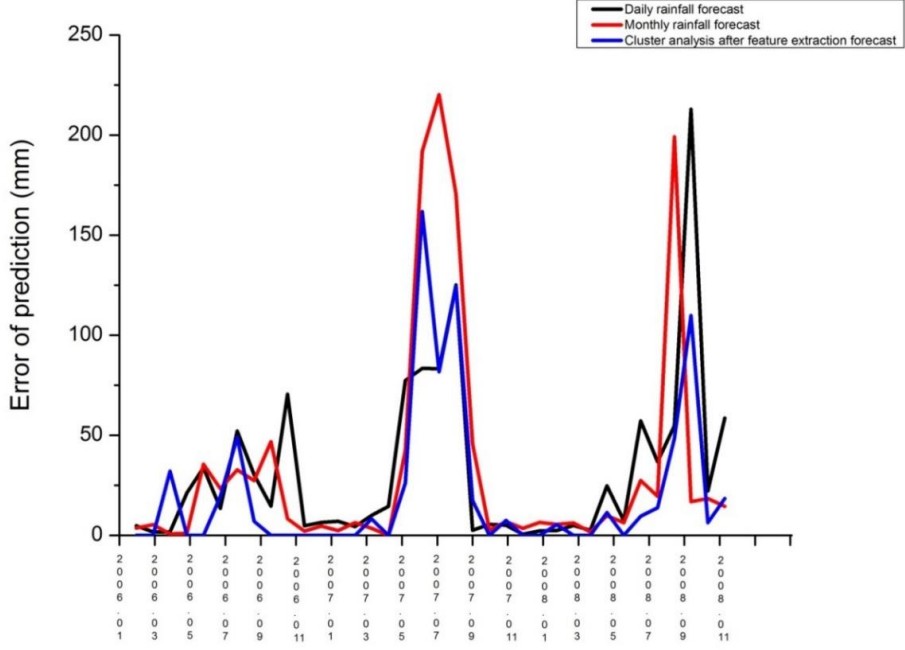

Fig.6. Comparison of the displacement prediction errors based on the 3 types of rainfall input



Natural Hazards
and Earth System
As can be seen from the figures and tables above, when using the original daily rainfall volume data, we have too much data for the neural network to process. The operation time of training is as high as $1.21 \times 109$. The neural network, unfortunately, has very limited capability to handle large volume of data. There are too many possible matching internal functions in the training stage. Therefore, we have the smallest mean squared error in the training stage but not the best prediction among the three methods.

In the second approach, when we use monthly average rainfall volume to forecast displacement, the volume of data to be processed is greatly reduced; but it is at the sacrifice of great reduction in rainfall features. In both the training and prediction stage, the results are the worst among the three approaches.

In the third method, when we use rainfall data after feature extraction we also have much less volume of data for the neural network to process, by comparison with using the first rainfall type; meanwhile, it is not at the sacrifice of great reduction in rainfall features when compared with the second approach. Although we have slightly higher mean squared error in the training stage, but the prediction results are the best among the three methods.

## 4 Result Discussion

We have used 2 years and 7 months data to train the BP neural network and made 3-year forecasting of the displacement of landslides (Figs. 5 and 6). The results showed some important features. First, by using the proposed feature extraction approach of the rainfall data the computational burden for forecasting was greatly reduced. Second, the comparison of the predicted and the observed displacement indicates that using the feature extraction approach has led less forecasting error than using other rainfall reduction methods (e.g., monthly or 60-day average). Moreover, one more interesting feature is noteworthy. From the prediction results (Fig. 5) we can see that the forecasting capability has no significant decay with the increase of time accumulation. The prediction of the displacement peak in the summer of 2008 is even more precise than the prediction of the peak in summer 2007. This fact may lead us to suspect that either there are other significant contributing factor(s) to the displacement peak in 2007; or there are more characteristics in the rainfall in summer 2007 that has not been essentially characterized by the current approach. After all, we can confidently state that the feature extraction approach is an important improvement in rainfall-landslide characterization process.

## 5 Conclusions

In this paper, we first analysed the characteristics of rainfall data, extracted the rainfall volume, rainfall duration and rainfall time characteristics for each single rainfall event. The amount of rainfall data is greatly reduced and the characteristics of rainfall data are substantially extracted. As the second step, the feature information of rainfalls was used in landslide displacement prediction. After feature extraction in rainfall data, we used the extracted features for the characteristic analysis and prediction to the Baishuihe landslide. We selected the basic method of BP neural network and applied it to three types of rainfall data include the characteristic value, the daily rainfall and monthly rainfall as input and forecast landslide displacement individually and compared the errors and efficiency. We can





reach a conclusion of that using the feature extracted rainfall input is superior to the rainfall types of daily rainfall
and monthly average rainfall in landslide displacement prediction.

**Acknowledgements**

This research was funded by the National Natural Sciences Foundation of China (Project Nos. 41302278, 41272377,
and 41272306), and the Fundamental Research Funds for National Universities, China University of Geosciences-
Wuhan (No. CUG120119). The authors are grateful to the Zhangjiachong Soil and Water Conservation Experiment
Station in Zigui County for providing the rainfall data. The first author wishes to thank the China Scholarship
Council for funding his visit to the University of Connecticut where this study was conducted.

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
