# Peer review of "Rainfall feature extraction using cluster analysis and its application on displacement prediction for a cleavage-parallel landslide in the Three-Gorges Reservoir area"

_Natural Hazards and Earth System Sciences, 2015_

## Short Comment (SC1) · 22 Jan 2016

Please note that the correct year of pubblication of the work the paper Melillo et al. is 2015. The correct reference is: Melillo M, Brunetti MT, Peruccacci S, Gariano SL, Guzzetti F. An algorithm for the objective reconstruction of rainfall events responsible for landslides. Landslides 12(2): 311-320, 2015. (http://link.springer.com/article/10.1007%2Fs10346-014-0471-3)

---

## Author Comment (AC1) · 22 Jan 2016

The last updated figure is for Fig 6. not Fig. 1.
[Figure]

Y-axis: Error of prediction(mm)

Legend:
- Daily rainfall forecast
- Monthly rainfall forecast
- Cluster analysis after feature extraction forecast

X-axis labels: 2006.01 2006.04 2006.07 2006.10 2007.01 2007.04 2007.07 2007.10 2008.01 2008.04 2008.07 2008.10

**Fig. 1.**

---

## Referee Comment (RC1) · Anonymous Referee #1 · 29 Feb 2016

GENERAL COMMENTS

The contribution "Rainfall feature extraction using cluster analysis and its application on displacement prediction for a cleavage-parallel landslide in the Three-Gorges Reservoir area" by Y. Liu M. and co-Authors is good and potentially publishable. The Authors present and employ cluster analysis (CA)-based feature analysis to rainfall data for rainfall feature extraction. This method extracts the most significant features of a rainfall sequence and greatly reduced rainfall data quantities. This approach is applied and validated to a data set acquired at a cleavage-parallel landslide in the Three-Gorges Reservoir area. The topic address scientific questions within the scope of NHESS.

The theoretical background is well-argued. Review of literature seems complete. The description of study area is sufficiently complete. The description of methodology and successive parts of paper are not well organized. Results, and discussion sections are very short compared to amount of work done. They should be widely increased. The readability of the whole paper is sufficient with a quite good English, which however can be improved. Overall, the work presents some carelessness and incompleteness. It can be published on NHESS journal only after a major revision.

SPECIFIC COMMENTS

I have some specific comments that should be addressed before the manuscript can be accepted for publication.

1) Section 2.1, "The relationship between rainfall and evaporation" (Page 3, Lines 82-92).

Authors should better explain if they are dealing with real or potential evapotranspiration. Moreover, some details on the calculation of evapotranspiration should be needed. Furthermore, how they passed from monthly to daily ET?

2) Section 2.2, "Statistics of rainfall by times" (Page 3, Lines 93-101).

Authors stated that "most studies carry out statistics analysis of rainfall based on precipitation per month or per day". But many papers are present in the literature in which statistical analysis are carried out using daily rainfall data. Authors should considerer these works.

3) Section 2.3, "The features of the rainfall data: rainfall volume, rainfall duration and rainfall time" (Page 4, Lines 102-126).

For a better clarity and understanding of the text, Authors should specify in detail how the values of the parameters used to evaluate the three indexes were chosen.

4) Section 2.4, "Clustering Analysis using the K-means clustering algorithm" (Page 4, Lines 127-135, Page 5, Lines 136-143).

The main methodology of the paper is represented by the application of the K-means clustering algorithm. All the used variables are shortly introduced and this leads to some misunderstandings. A figure with a flow chart would be very useful to understand the variables and the all the process.

5) Section 3.1, "Geological background and data collection" (Page 5, Lines 145-158).

Authors should better explain why they have chosen the ZG93 point. Is it representative for all the landslide body?

6) Section 3.2, "Feature analysis of rainfall data" (Page 5, Lines 159-169, Page 6, Lines 170-182, Page 7, Lines 184-187).

A column chart with the average monthly rainfall would be needed. Moreover, also an ECDF graph for duration and cumulated rainfall would be useful for analyzing differences.

7) Section 3.3, "Feature extraction of Rainfall data and Categorization results" (Page 7, Lines 189-203, Page 8, Lines 205-218, Page 9, Lines 219-237).

This paragraph is very confusing. The definition of the three indices are unclear. How Authors obtained the values for p1 and p2? What "scaling coefficient" means?

8) Section 3.4, "Prediction of landslide displacement with BP neural network" (Page 9, Lines 238-250, Page 10, Lines 253-265, Page 11, Lines 266-270, Page 12, Lines 271-282).

Several variables are introduced but no longer used in the following.

A sensitivity analysis, considering several validation periods (in addition to the one used in the work: 2006-2008) would be needed in order to evaluate the performance of the analysis.

9) Section 4, "Result Discussion" (Page 12, Lines 283-295)

This section is very short. Authors should better argue and comment the obtained results.

10) Section 5 "Conclusion" (Page 12, Lines 296-303, Page 13, Lines 304-305)

Poor conclusions. Authors should better explain the main findings and implications of their work.

TECHNICAL CORRECTIONS

Page 1, Line 30: Please rewrite better the following sentence "At present time".I suggest to use "At the present".

Page 1, Line 31: I suggest to change "is" with "are".

Page 2, Line 69: I suggest to change "Land slide" with "Landslide".

Page 3, Line 75: I suggest to define a variable for the cumulative rainfall. Please insert "*E (mm)*" and rewrite "cumulative rainfall *E* (mm)".

Page 3, Line 75: I suggest to define a variable for the average annual rainfall. Please insert "*MAP (mm)*" and rewrite "average annual rainfall *MAP* (mm)".

Page 3, Line 75: I suggest to define a variable for the monthly average of evaporation. Please insert "*MME (mm)*" and rewrite "monthly average of evaporation *MME* (mm).

Page 3, Line 85: I suggest to replace "mm/d" with "mmd$^{-1}$".

Page 3, Line 86: I suggest to change "day" with "days".

Page 4, Line 108: I suggest to define better the name of variables for the rainfall volume, rainfall duration and rainfall time

Page 5, Line 149: I suggest to change "140-m" with "140 m".

Page 5, Line 150: I suggest to change "600-m" with "600 m".

Page 5, Line 151: I suggest to change "700-m" with "700 m".

Page 5, Line 169: I suggest to replace "mm/d" with "mmd$^{-1}$" and please use the same number of decimal places. Please correct "4" with "4.0" and "6.26" with "6.3".

Page 6, Line 175: I suggest to replace "mm/d" with "mmd$^{-1}$" and please use the same number of decimal places. Please correct "1.28" with "1.3".

Page 5, Line 169: I suggest to replace "mm/d" with "mmd$^{-1}$" and please use the same number of decimal places. Please correct "4" with "4.0" and "6.26" with "6.3".

Page 6, Lines from 180 to 182: Please use the same format for the text.

Page 6, Line 180:I suggest to replace "N equals to 2" with "N = 2".

Page 6, Figure 1: Please use the same graphic element for represent the horizontal scale and North indicator symbol.

Page 7, Figure 2: Please use an appropriate format for the x-axes, please remove the ticks on the upper x-axes. Please use a better representation for the legend.

Page 7, Figure 2: I suggest to separate the values of Year/cumulated rainfall from graph with a new table.

Page 7, Figure 2: Please use the same number of decimal places.

Page 8, Figure 3: I suggest to use a 2D graph for represent the r, d variables, and a different scale of colours for represent the T value.

Page 8, Lines 211-212: I suggest to use a subscript index. Please change "C1" with "$C_1$", "C2" with "$C_2$", "C3" with "$C_3$" and "C4" with "$C_4$"

Page 8, Line 212: Numbers reported in the text " C4=(2.45, 4, 7.33)" do not always meet them reported in Figure 3. Please check.

Page 10, Figure 4: Please use the same format for all the graphs.

Page 10, Table 1:Please use a variables to report in table the three types of rainfall input data. Please use the same number of decimal places.

Page 11, Figure5, 6: Please use the same format for all the graphs. In particular, the authors use the same colours to represent the values of displacement and value of the prediction error relative to the three types of rainfall input data.

Page 11, Figure5, 6: I suggest to use a two q-q plots representation. The quantile-quantile or q-q plot is an exploratory graphical device used to check the validity of a distributional assumption for a data set.

---

## Referee Comment (RC2) · Anonymous Referee #2 · 25 Mar 2016

The manuscript deals with the effect of rainfall and its role on landslide deformation and failure. The authors carried out a feature extraction method for a rainfall data set and it was categorized by a cluster analysis. Rainfall indexes were computed for rainfall characteristics such as quantity, duration, and the number of raining days in a given period of time. The results were later applied and validated to a data set acquired at a cleavage-parallel landslide in the Three-Gorges Reservoir area. The landslide displacement prediction using neural networks for the rainfall input in the form of raw data, monthly rainfall, and feature extracted rainfall were benchmarked. The authors concluded that using the feature extracted rainfall method is best at predicting landslide

displacement compared to the other methods and at the same time the computational stress has been reduced significantly.

Although the topic is very interesting from a scientific and practical point of view, the manuscript presents some limitations, conceptual mistakes, technical errors and is sometimes confusing to read. Consequently, it is not suitable for in the present form. The paper must undergo major revisions for publication in NHESS.

The authors are strongly encouraged to review the paper in accordance to the high international standards of the NHESS Journal.

General comments:

- The authors should re-organize the paper to have a coherent scheme regarding the presentation of the work carried out. At the moment the manuscript contains a lot of relevant information but it is scattered and spread around the paper in a disorganized manner. The authors are encouraged to consolidate this information inside the relevant sections of the manuscript and to avoid unnecessary repetitions.

- The manuscript lacks the relevant references in the topic and in addition only 13 references are cited inside the document from the 44 listed in the reference list. It is highly recommended that the authors should carefully look into this.

- Regarding also references, the paper is lacking of an analysis of the important results and issues raised by other studies, in particular in context of the submitted paper. Discussion of the results obtained in the submitted manuscript should be made by comparing qualitatively and if possible quantitatively with the results obtained in referenced studies.

- Basic descriptions and concepts are not defined inside the manuscript, such as: cleavage-parallel landslide and BP neural networks (to name a few).

- It is recommended that the authors revise the manuscript all over again and find the suitable words, phrases, technical terms and definitions in proper English. It becomes

even more critical when the authors pretend to describe the methodology. Detailed comments:

- The abstract should be improved. Some statements like the one in Line 14 and Line 16 are misleading and should be rephrased.

- The introduction is lacking relevant references. This introductory part should be re-arranged in a way that the references are supporting the stated comments (i.e. Line 31).

- The second paragraph of the Introduction should be fully rephrased and a better summary of the past studies should be carried out by the authors.

- In Line 33, the authors should explain in detail how this is difficult (with supporting references) and how their method is an improvement for this.

- In Line 52, Kurtz et al. 2014 is incorrectly referenced in the manuscript. This work is not relevant to this paper and is based on other approach for feature extraction. The authors should use another reference or delete this one.

- In Line 55, the authors should use the correct references to support the sentence.

- Line 70 should describe briefly what a BP neural network is.

- In Line 74, a description of this type of landslide should be included.

- The Methodology section should be fixed and improved. A better description should be included in order to make the reader understand better the approach used.

- Line 85 should include references.

- Line 89 to Line 92, the authors should explain in detail the reason to use an average daily evaporation? Is it because of lack of data or is it a common practice?

- Line 94 to Line 96, the authors should rephrase the statement and add the relevant reference.

- Line 109 to 111 should explain, why the authors use this approach and why.

- Line 121 to 124 needs to be fixed and rephrased. It is not understandable, what the authors mean with high cohesion and low coupling in this context.

- In line 128, the authors mention, that the K-means algorithm is the most used clustering algorithm. In what context and explain the purposes.

- Line 128 to 135 needs relevant references

- Line 136. Also the sample selection affects the final results. The authors should elaborate in this respect also.

- Line 148: What does it means that it is shaped like stairs to the Yangtze River?

- Line 153 to 158: the authors should explain in detail, why are they using that control point and why is that significant. They also should include the figure of the profile and where the point is located, inside figure 1.

- Line 158: the authors should reflect on adding a new point or several others as the landslide movement is not uniform and it will add value to the validation. This will definitely have a big influence on the final results.

- Line 160: the authors should include the Zigui County in Hubei Province inside a map to understand the actual location and to understand if this will affect the final results as rainfall is spatially correlated.

- Line 171, in figure 1, the letters should go first, then the legend.

- Line 175. Using average interpolation for a long period can be misleading as the warmer days without rain will tend to influence the sample. Can the authors elaborate why not only using the evaporation related to the rainy days?

- Figure 1: the red square in the smaller map is barely visible.

- Line 196: the authors should explain how this coefficient is derived.

- Line 203: the authors should explain how this optimal point was achieved.

- Line 190 to 203: It is unclear inside the manuscript, why the aforementioned coefficients were used and its relevance to the results. Can similar results be achieved using other techniques of normalization?

- Figure 3: Check on the r-axis. Point C3 is out of the range.

- Line 211: explain briefly the principle of maximum dissimilarity.

- Line 214 to 219: The authors should use the relevant references.

- Line 219: What is the stability criterion?

- Line 228: The authors should explain how they were coming to this conclusion.

- Line 248 to 249: Use relevant reference.

- Figure 5 and 6: Both daily forecasts and cluster analysis seem to give reasonable results. Is there a better way to show the improvement of the cluster analysis (a success rate approach)?

- Figure 5 and 6: One of the main advantages of the cluster analysis is based on the improvement of the computational effort. Can the authors further elaborate; is this improvement really worth it (as there will be also computation involved on the K-means algorithm)?

- The authors should discuss in the conclusion section, if this approach can be improved by selecting other type of clusters. Is there a possibility to perform a sensitivity analysis based on this?

- The conclusions are mostly a repetition of the text inside the manuscript. The conclusion section should be fixed and rephrased.

---

## Author Comment (AC2) · 17 Apr 2016

Dear NHESS Editors and Referees,

First of all, we would like to express our sincere appreciation for your very detailed and constructive comments and suggestions.

Next, in a sequence, we would like to respond to your comments in a point to point manner so that hopefully all the questions can be answered or clarified. All the answers and responses are in red.

GENERAL COMMENTS

The Authors present and employ cluster analysis (CA)-based feature analysis to rainfall data for rainfall feature extraction. This method extracts the most significant features of a rainfall sequence and greatly reduced rainfall data quantities. This approach is applied and validated to a data set acquired at a cleavage-parallel landslide in the Three-Gorges Reservoir area. The topic address scientific questions within the scope of NHESS.

A: Thanks for your encouraging words.

The theoretical background is well-argued. Review of literature seems complete. The description of study area is sufficiently complete. The description of methodology and successive parts of paper are not well organized. Results and discussion sections are very short compared to amount of work done. They should be widely increased. The readability of the whole paper is sufficient with a quite good English, which however can be improved. Overall, the work presents some carelessness and incompleteness. It can be published on NHESS journal only after a major revision.

A: We have closely followed your detailed comments, by:

1) The description of methodology are re-organized;
2) Results and discussion sections are expanded to include more discussions.
3) We have carefully checked the grammar and spellings, include the numerical values; we tried to eliminate the incompleteness to our best knowledge.

SPECIFIC COMMENTS

I have some specific comments that should be addressed before the manuscript can be accepted for publication.

1) Section 2.1, "The relationship between rainfall and evaporation" (Page 3, Lines 82-92).

Authors should better explain if they are dealing with real or potential evapotranspiration. Moreover, some details on the calculation of evapotranspiration should be needed. Furthermore, how they passed from monthly to daily ET?

A: The evapotranspiration is real observation data; but they are in monthly value. The daily ET value has been gotten simply by using the monthly value divided by number of days in that month. Due to the lack of other supplementary observations such as the temperature, moisture, wind speed, albedo etc., this is the best way we can estimate the daily ET. We have added the sentence in Page 3 Lines 95 – 97 to point out how the daily ET is calculated, as

"Nevertheless, we would like to point out that the daily evaporation value is calculated by simple division of the monthly value with the number of days in that month. This is the most practical way we can do, due to lack of more detailed supplementary meteorological observations in this area."

2) Section 2.2, "Statistics of rainfall by times" (Page 3, Lines 93-101).

Authors stated that "most studies carry out statistics analysis of rainfall based on precipitation per month or per day". But many papers are present in the literature in which statistical analysis are carried out using daily rainfall data. Authors should considerer these works.

A: Thanks for Your constructive suggestion to make the studies more comprehensively reviewed. We have added the followings references (also added in the reference list) on statistical analysis using daily data, in Section 2.2, after the first sentence:

"For example, Crozier and Eyles (1980) used daily rainfall and established thresholds to compare terrain sensitivity and to assess the occurrence probability of landslide. Using daily rainfall data from Kuala Kenderong and Kg. Jeli along the Gerik-Jeli Highway, Lateh et al. (2013) analysed the correlation of landslide events and rainfall precipitation. The rainfall induced landslides was investigated by applying the cumulative rainfall method which comprises the reconstruction of absolute antecedent rainfall for 20 landslide events."

3) Section 2.3, "The features of the rainfall data: rainfall volume, rainfall duration and rainfall time" (Page 4, Lines 102-126).

For a better clarity and understanding of the text, Authors should specify in detail how the values of the parameters used to evaluate the three indexes were chosen.

A: The values we chose are based on numerical sensitivity test for getting high coherence and low coupling of the rainfall-slide events. We have clarify it in Section 2.3.

4) Section 2.4, "Clustering Analysis using the K-means clustering algorithm" (Page 4, Lines 127-135, Page 5, Lines 136-143).

The main methodology of the paper is represented by the application of the K-means clustering algorithm. All the used variables are shortly introduced and this leads to some misunderstandings. A figure with a flow chart would be very useful to understand the variables and the all the process.

A: Section 2.4 has been re-written, a new figure to show the flow chart of the data processing procedures (also show below) is added.

[Figure]

Figure 1: The flow chart of the K-means algorithm.

5) Section 3.1, "Geological background and data collection" (Page 5, Lines 145-158).

Authors should better explain why they have chosen the ZG93 point. Is it representative for all the landslide body?

A: The rationale, along with the new Figure, is added as:

"The selection of ZG93 is based on: 1) It is roughly located at the center of the Baishuihe Landslide so that it is the most unlikely point to be contaminated by false alarm or local signals generated by boundary effect in those monitoring points close to landslide flanks; 2) Observational facts, as shown as the red curve and triangles in Figure 3 below, support our selection for the fact that it is sensitive enough to catch the subtle displacement in the early stage of the monitoring period (prior to the end of 2007) on one hand; and behaved as the average of all the point after rapid change occurred in May 2007 on the other hand."

[Figure]

Figure 3: The cumulative displacement of monitoring points in the Baishuihe Landslide.

6) Section 3.2, "Feature analysis of rainfall data" (Page 5, Lines 159-169, Page 6, Lines 170-182, Page 7, Lines 184-187).

A column chart with the average monthly rainfall would be needed. Moreover, also an ECDF graph for duration and cumulated rainfall would be useful for analyzing differences.

A: The suggested column chart with the average monthly rainfall is added as Fig. 5. and the ECDF graphs for duration and cumulative rainfall are added as Fig. 6. Thanks for Your constructive comments.

[Figure]

Figure 5. The average monthly rainfall column chart for the period of 2003-2008.

[Figure]

Figure 6. The ECDF plots of the cumulative rainfall and the duration for rainfall events.

7) Section 3.3, "Feature extraction of Rainfall data and Categorization results" (Page 7, Lines 189-203, Page 8, Lines 205-218, Page 9, Lines 219-237).

This paragraph is very confusing. The definition of the three indices are unclear. How Authors obtained the values for p1 and p2? What "scaling coefficient" means?

A: This section has been re-written. The 3 indices are definite first (r, d, and T), before they are discussed. The scaling coefficients are simply for best separation of the classes in cluster analysis, so that they could be any value. They are simply for better visual effect when plot out.

8) Section 3.4, "Prediction of landslide displacement with BP neural network" (Page 9, Lines 238-250, Page 10, Lines 253-265, Page 11, Lines 266-270, Page 12, Lines 271-282).

Several variables are introduced but no longer used in the following.

A: We have taken the advice. These variables are further illustrated and used in Table 2.

A sensitivity analysis, considering several validation periods (in addition to the one used in the work: 2006-2008) would be needed in order to evaluate the performance of the analysis.

A: Actually, the selection of train data and test data is not strict. However, a good performance of BP neural network requires enough train data. In this paper, less than half of data is selected as train data and good results are obtained.

9) Section 4, "Result Discussion" (Page 12, Lines 283-295)

This section is very short. Authors should better argue and comment the obtained results.

A: The conclusion is substantially re-written and expanded.

10) Section 5 "Conclusion" (Page 12, Lines 296-303, Page 13, Lines 304-305)

Poor conclusions. Authors should better explain the main findings and implications of their work.

A: The conclusion part has been re-written and expanded to re-cap the main findings and point out the direction of future study along this line of thinking.

TECHNICAL CORRECTIONS

1. Page 1, Line 30: Please rewrite better the following sentence "At present time". I suggest to use "At the present". (changed)

2. Page 1, Line 31: I suggest to change "is" with "are". (changed)

3. Page 2, Line 69: I suggest to change "Land slide" with "Landslide". (changed)

4. Page 3, Line 75: I suggest to define a variable for the cumulative rainfall. Please insert "E (mm)" and rewrite "cumulative rainfall E (mm)". (changed)

5. Page 3, Line 75: I suggest to define a variable for the average annual rainfall. Please insert "MAP (mm)" and rewrite "average annual rainfall MAP (mm)". (changed)

6. Page 3, Line 75: I suggest to define a variable for the monthly average of evaporation. Please insert "MME (mm)" and rewrite "monthly average of evaporation MME (mm). (changed)

7. Page 3, Line 85: I suggest to replace "mm/d" with "mmd-1". (changed)

8. Page 3, Line 86: I suggest to change "day" with "days". (changed)

9. Page 4, Line 108: I suggest to define better the name of variables for the rainfall volume, rainfall duration and rainfall time (not changed, since they are defined clear enough)

10. Page 5, Line 149: I suggest to change "140-m" with "140 m". (changed)

11. Page 5, Line 150: I suggest to change "600-m" with "600 m". (changed)

12. Page 5, Line 151: I suggest to change "700-m" with "700 m". (changed)

13. Page 5, Line 169: I suggest to replace "mm/d" with "mmd-1" and please use the same number of decimal places. Please correct "4" with "4.0" and "6.26" with "6.3". (changed)

14. Page 6, Line 175: I suggest to replace "mm/d" with "mmd-1" and please use the same number of decimal places. Please correct "1.28" with "1.3". (changed)

15. Page 5, Line 169: I suggest to replace "mm/d" with "mmd-1" and please use the same number of decimal places. Please correct "4" with "4.0" and "6.26" with "6.3". (changed)

16. Page 6, Lines from 180 to 182: Please use the same format for the text. (changed)

17. Page 6, Line 180:I suggest to replace "N equals to 2" with "N = 2". (changed)

18. Page 6, Figure 1: Please use the same graphic element for represent the horizontal scale and North indicator symbol. (changed)

19. Page 7, Figure 2: Please use an appropriate format for the x-axes, please remove the ticks on the upper x-axes. Please use a better representation for the legend. (changed)

20. Page 7, Figure 2: I suggest to separate the values of Year/cumulated rainfall from graph with a new table. changed)

21. Page 7, Figure 2: Please use the same number of decimal places. (changed)

22. Page 8, Figure 3: I suggest to use a 2D graph for represent the r, d variables, and a different scale of colours for represent the T value.

    A: We still believe that the 3D plot is a better way to express the relationships among the parameters. If a 2D plot is adopted, it will result in a wrong impression that T is the function of r and D; actually, it is not and is independent of r and d.

23. Page 8, Lines 211-212: I suggest to use a subscript index. Please change "C1" with "C1", "C2" with "C2", "C3" with "C3" and "C4" with "C4" (changed)

24. Page 8, Line 212: Numbers reported in the text "C4 = (2.45, 4, 7.33)" do not always meet them reported in Figure 3. Please check.

    A: Actually it might be caused by the visual distortion when it is projected into a transparent 3D coordinate.
    Page 10, Figure 4: Please use the same format for all the graphs. (changed)

25. Page 10, Table 1: Please use a variables to report in table the three types of rainfall input data. Please use the same number of decimal places.

    A: The decimal places have been unified.

26. Page 11, Figure5, 6: Please use the same format for all the graphs. In particular, the authors use the same colors to represent the values of displacement and value of the prediction error relative to the three types of rainfall input data.

    A: The may be caused by the pdf file has not truly reflect the figure expression in MS Word. See the revised draft for a better clarification.

27. Page 11, Figure5, 6: I suggest to use a two q-q plots representation. The quantile-quantile or q-q plot is an exploratory graphical device used to check the validity of a distributional assumption for a data set.

    A: A q-q plot for landslide displacement prediction based on 3 types of rainfall input and corresponding analysis has been added as follows.

[Figure]

Figure 11: The q-q plot for landslide displacement prediction based on 3 types of rainfall input.

"The q-q plot shown in Fig. 11 is an exploratory graphical expression used to check the validity of a distributional assumption for data sets. It is employed for analyzing the relationship between observed displacement data and the predictions with three types of rainfall input. If the observed and the predicted data sets have the same distribution, the fitted line in the q-q plot will approach y=x. As can be seen from Fig. 11, the fitted curve of the data points from the prediction with extracted rainfall feature is closer to the line y=x with slope of 1; while the prediction with monthly total rainfall is overestimated and the prediction with daily rainfall of 60 days is underestimated. It indicates that the extracted rainfall feature represents real rainfall better than daily rainfall of 60 days and monthly rainfall in landslide displacement prediction."

We sincerely appreciate the detailed and constructive comments and suggestions from Referee #1.

---

## Author Comment (AC3) · 17 Apr 2016

Dear NHESS Editors,

First of all, we would like to express our sincere appreciation for your very detailed and constructive comments and suggestions.

Next, in a sequence, we would like to respond to your comments in a point to point manner so that hopefully all the questions can be answered or clarified. All the answers and responses are in red.

GENERAL COMMENTS

The manuscript deals with the effect of rainfall and its role on landslide deformation and failure. The authors carried out a feature extraction method for a rainfall data set and it was categorized by a cluster analysis. Rainfall indexes were computed for rainfall characteristics such as quantity, duration, and the number of raining days in a given period of time. The results were later applied and validated to a data set acquired at a cleavage-parallel landslide in the Three-Gorges Reservoir area. The landslide displacement prediction using neural networks for the rainfall input in the form of raw data, monthly rainfall, and feature extracted rainfall were benchmarked. The authors concluded that using the feature extracted rainfall method is best at predicting landslide displacement compared to the other methods and at the same time the computational stress has been reduced significantly.

Although the topic is very interesting from a scientific and practical point of view, the manuscript presents some limitations, conceptual mistakes, technical errors and is sometimes confusing to read. Consequently, it is not suitable for in the present form. The paper must undergo major revisions for publication in NHESS.

The authors are strongly encouraged to review the paper in accordance to the high international standards of the NHESS Journal.

General comments: - The authors should re-organize the paper to have a coherent scheme regarding the presentation of the work carried out. At the moment the manuscript contains a lot of relevant information but it is scattered and spread around the paper in a disorganized manner. The authors are encouraged to consolidate this information inside the relevant sections of the manuscript and to avoid unnecessary repetitions.

- The manuscript lacks the relevant references in the topic and in addition only 13 references are cited inside the document from the 44 listed in the reference list. It is highly recommended that the authors should carefully look into this.

- Regarding also references, the paper is lacking of an analysis of the important results and issues raised by other studies, in particular in context of the submitted paper. Discussion of the results obtained in the submitted manuscript should be made by comparing qualitatively and if possible quantitatively with the results obtained in referenced studies.

- Basic descriptions and concepts are not defined inside the manuscript, such as: cleavage-parallel landslide and BP neural networks (to name a few).

- It is recommended that the authors revise the manuscript all over again and find the suitable words, phrases, technical terms and definitions in proper English. It becomes even more critical when the authors pretend to describe the methodology. Detailed comments:

A: We sincerely appreciate your constructive, detailed comments to the manuscript.

1) We have thoroughly revised the manuscript to make it more precise, consistent, and clear.

2) We have thoroughly checked all the concepts are clearly defined.

3) All the references to make sure all the cited papers are listed, and all the listed have been cited. The abstract and conclusion are re-written.

We have the one-to-one responses to the comments as follows.

1) The abstract should be improved. Some statements like the one in Line 14 and Line 16 are misleading and should be rephrased.

A: The abstract and conclusion are re-written as:

"In this paper, a feature extraction method using a cluster analysis (CA) is employed for the analysis of rainfall data. With this approach we effectively revealed the most significant features contained in a rainfall sequence and greatly reduced the burden for processing large amount of rainfall data. Meanwhile, it greatly improves the spectrum of usefulness of rainfall data."

2) The introduction is lacking relevant references. This introductory part should be rearranged in a way that the references are supporting the stated comments (i.e. Line 31).

3) The second paragraph of the Introduction should be fully rephrased and a better summary of the past studies should be carried out by the authors.

4) In Line 33, the authors should explain in detail how this is difficult (with supporting references) and how their method is an improvement for this.

A: The Introduction has been substantially re-written, and all the comments from both Referees are incorporated, and more references are added.

5) In Line 52, Kurtz et al. 2014 is incorrectly referenced in the manuscript. This work is not relevant to this paper and is based on other approach for feature extraction. The authors should use another reference or delete this one.

A: Thanks for pointing out this mistake. Kurtz et al 2014 has been deleted from the text and the reference list.

6) In Line 55, the authors should use the correct references to support the sentence.

A: The following references are added and citied here to support the claim:

Wang MJ, Shen, JH. Rainfall Landslide and Debris Flow Intergrowth Relationship in Jiangjia Ravine [J]. Journal of Mountain Science, 2011, 8(4):603-610.

Finlay PJ. The relationship between the probability of landslide occurrence and rainfall [J]. Canadian Geotechnical Journal, 1997, 34(6):811-824.

Gariano SL, Brunetti MT, Iovine G, Melillo M, Peruccacci O, Terranova C, Vennari F, Guzzetti F. Calibration and validation of rainfall thresholds for shallow landslide forecasting in Sicily, southern Italy[J]. Geomorphology, 2015, 228(1):653-665.

7) Line 70 should describe briefly what a BP neural network is.

A: The short introduction of BP neural network method is added as the second paragraph of the subsection 3.4:

"The back propagation (BP) network is a kind of multilayer feedforward neural network. It is a widely tested and validated error back propagation algorithm. The network consists of an input layer, a number of hidden (middle) layers and an output layer. Based on Kolmogorov's theorem, a three layer BP

neural network can achieve approximation for any arbitrary nonlinear functions, so that we choose BP neural network to carry out this quantitative examination."

8) In Line 74, a description of this type of landslide should be included.

A: Cleavage-parallel landslide is defined at the start of Methodology.

9) The Methodology section should be fixed and improved. A better description should be included in order to make the reader understand better the approach used.

A: The Methodology part has been revises. This is similar to the comments from Referee #1.

10) Line 85 should include references.

A: The reference of Wu is added and cited:

Wu H. Monitoring and theoretical analysis of rainfall infiltration of Huangtupo landslide in the three gorges reservoir. China University of Geosciences for the Master Degree of Engineering, 2014.

11) Line 89 to Line 92, the authors should explain in detail the reason to use an average daily evaporation? Is it because of lack of data or is it a common practice?

A: This is the same comments from Referee #1, and we have modified the text to explain our approach.

12) Line 94 to Line 96, the authors should rephrase the statement and add the relevant reference.

A: This opening sentence has been modified, and the relevant reference are added:

Bui, D., Pradhan, B., Lofman, O., Revhaug, I. and Dick, Ø. B.: Regional prediction of landslide hazard using probability analysis of intense rainfall in the Hoa Binh province, Vietnam. Natural hazards, 66(2), 707-730, 2013.

Du J, Yin K and Lacasse S.: Displacement prediction in Colluvial landslides, three Gorges reservoir, China [J]. Landslides, 10(2): 203-218, 2013.

13) Line 109 to 111 should explain, why the authors use this approach and why.

A: The selection of daily average, duration and the contiguous days of rainfall is based on searching through a large quantity of previous studies. And the analysis results our supported support our selection by displaying significant correlation between the selected rainfall parameters and the deformation behavior.

14) Line 121 to 124 needs to be fixed and rephrased. It is not understandable, what the authors mean with high cohesion and low coupling in this context.

A: This has been modified. Thanks for the comments.

15) In line 128, the authors mention, that the K-means algorithm is the most used clustering algorithm. In what context and explain the purposes.

A: The applicability and maturity of the K-means method is emphasized. The whole paragraph has been re-written.

16) Line 128 to 135 needs relevant references

A: The citations are added:

Steinley D: K-means clustering: A half-century synthesis [J]. British Journal of Mathematical and Statistical Psychology, 59(1): 1-34, 2006.

Hartigan J. A., and Wong M. A.: A K-means clustering algorithm, Applied Statistics, 28, 100-108, doi: 10.2307/2346830, 2013.

17) Line 136. Also the sample selection affects the final results. The authors should elaborate in this respect also.

A: In our study the sampling selection is irrelevant since we have used all the 211 samples of the rainfall events.

18) Line 148: What does it means that it is shaped like stairs to the Yangtze River?

A: Terrace is the right geological term to be used. Now this part is changed to:

"It is a single, north-facing, inclined cleavage-parallel slope on the Yangtze River terrace."

19) Line 153 to 158: the authors should explain in detail, why are they using that control point and why is that significant. They also should include the figure of the profile and where the point is located, inside figure 1.

A: This is the same comments from Referee #1, has been answers and revised in the text.

20) Line 158: the authors should reflect on adding a new point or several others as the landslide movement is not uniform and it will add value to the validation. This will definitely have a big influence on the final results.

A: The primary purpose of this paper is to examine the effectiveness of the proposed clustering analysis approach. We will continue the work to include more cases by using data in more observation points. Thanks for your valuable suggestion.

21) Line 160: the authors should include the Zigui County in Hubei Province inside a map to understand the actual location and to understand if this will affect the final results as rainfall is spatially correlated.

A: The map has been modified with an inset to indicate the location of Zigui County.

[Figure]

Figure 2. (a) The location of the Baishuihe Landslide (the west most red open square) in the Three Gorges Reservoir area; (b) The locations of the GPS benchmarks (the red and magenta solid squares) for displacement monitoring in the Baishuihe Landslide; (c) The vertical geological cross-section of the Baishuihe Landslide along Profile I.

22) Line 171, in figure 1, the letters should go first, then the legend.
A: This has been revised in the figure 2.

23) Line 175. Using average interpolation for a long period can be misleading as the warmer days without rain will tend to influence the sample. Can the authors elaborate why not only using the evaporation related to the rainy days?
A: This is the same comments from Referee #1, has been answers and revised in the text.

24) Figure 1: the red square in the smaller map is barely visible.
A: This is the same comments from Referee #1, has been answers and revised in the text and the Fig. 2.

25)
Line 196: the authors should explain how this coefficient is derived.

Line 203: the authors should explain how this optimal point was achieved.

Line 190 to 203: It is unclear inside the manuscript, why the aforementioned coefficients were used and its relevance to the results. Can similar results be achieved using other techniques of normalization?

A: This is answered as the reply to Referee #1's Comments # 7).

26) Figure 3: Check on the r-axis. Point C3 is out of the range.

A: The figure has been modified to correct this imperfection.

[Figure]

Figure 7. The classified rainfall types based on cluster analysis: I: sporadic rainfall; II: long-duration rainfall; III: short-duration storms; and IV: long-duration intermittent rainfall.

27) Line 211: explain briefly the principle of maximum dissimilarity.

A: This is best illustrated by using the Eq. 2 shown in

$$D = \sum_{i=1}^{k} \sum_{j=1}^{k} (x_i - x_j)^2 \qquad (2)$$

In Eq. 2, D is the total distance (dissimilarity) the k data points share. The larger the D is, the higher probability that these k data points are in the different classes.

28) Line 219: What is the stability criterion?

A: In the revised version we have clearly defined the criterion for stopping further cluster classification.

29) Line 228: The authors should explain how they were coming to this conclusion.

This part has been modified as:

"Rainfall volume is the most important factor in causing the variations of displacement in the

cleavage-parallel landslide (Gariano et al., 2015). Therefore, after categorization, we use rainfall volume as the feature for extraction, taking the total rainfall volume in the same category as the feature of that particular category."

30)  Line 248 to 249: Use relevant reference.

A: The number of nodes for the hidden layers in BP neural networks is determined empirically. To make it clear, the description of this part is revised as:

"The number of nodes for the hidden layer n1 in BP neural networks is determined empirically with

$$n_1 = \sqrt{n + m} + a$$

where n is the number of nodes for the input layer; m is the number of nodes for the output layer; and a is a constant, which is set to be 2 in this work. The results of $n_1$ are 9, 4, and 5 in the BP neural networks for the three rainfall categories. The information of the number of nodes in different layers, along with the errors in prediction, are shown in Table 2."

31) Figure 5 and 6: Both daily forecasts and cluster analysis seem to give reasonable results. Is there a better way to show the improvement of the cluster analysis (a success rate approach)?

A: The selection of which clustering method to be used is not the theme of this paper. However, the K-means method is a matured clustering method. This is the reason we have chosen it as the method to be used in this paper.

32) Figure 5 and 6: One of the main advantages of the cluster analysis is based on the improvement of the computational effort. Can the authors further elaborate; is this improvement really worth it (as there will be also computation involved on the K-means algorithm)?

A: To reduce the computational burden is only one of the advantages of using clustering analysis. Besides the factor of reducing computational intensity, it also improves the accuracy of the prediction. It provides another way to preprocess the rainfall data.

33) The authors should discuss in the conclusion section, if this approach can be improved by selecting other type of clusters. Is there a possibility to perform a sensitivity analysis based on this?

A: The selection of which clustering method to be used is not the theme of this paper. However, the K-means method is a matured clustering method. This is the reason we have chosen it as the method to be used in this paper. This is the similar answer for the question of the above regarding Figs 9 and 10.

It is hard to conduct any sensitivity analysis, since we only have the monthly deformation data. It is hard to discuss which clustering approach of the rainfall is more sensitive against the relatively coarse in temporal monthly deformation.

34) The conclusions are mostly a repetition of the text inside the manuscript. The conclusion section should be fixed and rephrased.

A: This is the same comments from Referee #1, has been answers and revised in the text.

Sincerely thanks for your detailed and constructive comments and suggestions.